# The Effect of Boredom on College Students’ Meaning in Life: A Longitudinal Mediation Model

**DOI:** 10.3390/ijerph191912255

**Published:** 2022-09-27

**Authors:** Xushan Li, Xiaoming Jia

**Affiliations:** 1Mental Health Education and Counseling Center, Beijing Institute of Technology, Beijing 100081, China; 2School of Humanities and Social Sciences, Beijing Institute of Technology, Beijing 100081, China

**Keywords:** boredom, meaning in life, college students

## Abstract

Boredom is a common emotional experience in daily life adversely affecting individual physical health, mental health, and social functioning. Therefore, it is an important issue to improve the quality of life by ameliorating individuals’ state of boredom. We used a longitudinal research approach. First, we tested 728 participants with the Multidimensional State Boredom Scale (MSBS). Then, after 3 months, the participants filled out the Cognitive Flexibility Inventory (CFI), Perceived Social Support Scale (PSSS), and Meaning of Life Questionnaire (MILQ); 715 valid questionnaires were obtained. Results showed that cognitive flexibility played a mediating role between boredom and the presence of meaning in life, but social support did not. The total effect of boredom on the presence of meaning in life was significant. Cognitive flexibility and social support played a mediating role between boredom and the search for meaning in life, respectively, but the overall effect of boredom on the search for meaning in life was not significant. This study found that boredom has different effects on the presence of meaning in life and the search for meaning in life. It can improve individuals’ sense of meaning in life by reducing boredom and improving cognitive flexibility and social support.

## 1. Introduction

### 1.1. Boredom and Effects

Previous research has mainly interpreted boredom in terms of emotional, cognitive, physiological, and motivational aspects. Although a unified conclusion has not been reached, it is universally agreed that boredom is an unpleasant emotional experience [1] that is associated with factors such as external stimuli, attentional cognition, physiological arousal [2,3], etc. Accordingly, we considered boredom a negative emotional state that arises because of an individual’s inability to concentrate and loss of interest in the current activity. Not only does boredom arouse trivial and transient dissatisfaction, but prolonged boredom can become increasingly distressing, leading to a range of psychological, social, and health problems. According to the stability of boredom across time and contexts, boredom can be divided into trait and state boredom [4]. Many studies focused more on trait boredom, while those on state boredom have not received extensive attention [5]. Therefore, this study focuses on state boredom. State boredom refers to the transient experience of boredom that individuals experience in a specific context, which is a conscious and subjective feeling mostly caused by the repetitive monotony of stimuli in the environment or a lack of engagement or involvement [6].

Boredom is a negative emotion in which individuals experience a loss of attention, lack of interest, poor self-control, and reduced well-being, which in turn affects their perception of their current situation and experience of the meaning of life [7,8]. In most studies that have examined the relationship between the meaning of life and boredom [9], boredom was not taken as an antecedent variable. 

### 1.2. Two Dimensions of Meaning in Life

Meaning in life is defined as people understanding and appreciating life, discovering the importance of their own lives, and recognizing the purpose and mission of life [10]. On the basis of research on the connotation of meaning in life, Steger et al. proposed a two-dimensional structure that indicated that individual meaning in life included not only the feeling that their lives are meaningful but also the pursuit of meaning, as in the presence of meaning in life and the search for meaning. The former is the degree to which individuals feel that their lives are meaningful, purposeful, and valuable, which is a core belief system; the latter is the intensity with which individuals invest in building or expanding their understanding of meaning in life [11]. The presence of meaning reflects the result or state of an individual’s discovery of meaning, while the search for meaning reflects the process of an individual’s pursuit of meaning in life [12]. Empirical studies have found a nonsignificant or low−level negative correlation between the presence and search for meaning [11,13]. They are also examined as two separate variables in most of the studies [14].

### 1.3. Intrinsic and Extrinsic Resources

The Five “A”s of Meaning Maintenance Model focuses on the intrinsic resources of individuals and theorizes on the mechanisms by which boredom affects meaning in life. It proposes that assimilation and accommodation, important concepts of Piaget’s cognitive development theory, are among the compensation strategies for individuals’ loss of meaning and arousal of disgust. Although the functions of the two are different, assimilation and accommodation both enrich the existing schemas of individuals and promote the development of cognitive structures. Piaget believes that to achieve adaptation, individuals need to constantly adjust their cognitive structures due to environmental constraints as a way to obtain a balanced process of internal cognition and external environment [15]. 

Cognitive flexibility is an important part of executive functioning as an individual’s ability to adopt flexible cognitive strategies, which reflects cognitive adaptability and plays an important role in individual ability development and environmental adaptation [16]. Martin et al. argued that cognitive flexibility originates from Piaget’s research on children’s cognitive development. This refers to an individual’s ability to realize the existence of different solutions in problem situations and adapt to new situations in a flexible manner to achieve specific goals. Individuals with a lack of cognitive flexibility usually show characteristics of being fixed, rigid, and unable to adapt to some changed situations or goals [17]. From this perspective, cognitive flexibility can be used as a manifestation of the level of assimilation and accommodation, which is the ability of individuals to freely adjust their cognition in response to different stimuli or environmental changes.

To cope with boredom, individuals must mobilize their intrinsic resources. Cognitive flexibility, as a functional expression of assimilation and accommodation, can help individuals recognize and understand dilemmas from different perspectives and solve problems in multiple ways [18], thus enabling them to mediate the relationship between boredom and meaning in life. However, Levenson et al. argued that emotions are both cognitive organizers and disruptors [19]. In terms of resource allocation theory, when threatening stimuli that induce negative emotions are present, individuals’ attentional resources are heavily occupied. As the attentional scope becomes narrower, thinking tends to become stereotypically passive when cognitive tasks with high demands on attentional resources appear [20]. From this perspective, boredom as a negative emotion may also prevent individuals from adopting flexible and effective cognitive strategies, which in turn may affect the presence of and research on meaning in life. Therefore, the specific pathway role of cognitive flexibility between boredom and meaning in life needs to be verified by further research.

Although the Five “A”s Meaning Maintenance Model focuses on intrinsic resources and theorizes on the mechanisms by which boredom affects the meaning in life, the role of extrinsic resources remains to be explored. Developmental resource theory suggests that adolescents cannot develop without intrinsic and extrinsic resources. Social support is an important extrinsic resource for individuals and is closely related to psychological well−being. Cassel argued that social support has a protective function and can act as a buffer or protector when individuals are under stress [21]. Moreover, Cobb believed that it is an interpersonal information exchange in which individuals perceive information related to the self, such as concern, respect, and needs for communication and belonging [22]. In the current study, social support was considered to be the process through which individuals interact with others and social organizations, receive material or moral help, feel cared for and respected, and thus improve their life resilience. At the same time, social support, as a manifestation of social relationships, is an important source of meaning in life [23,24]. When people report the sources of meaning in life, they often mention lovers and friends, which are parts of social support. Likewise, family is also an important source, as is shown in research that college students with a harmonious family atmosphere and well−functioning families have a higher sense of meaning in life [25]. Bored individuals are often accompanied by tendencies toward anxiety and depression, which are more likely to trigger problems such as interpersonal sensitivity. Boredom may affect their interpersonal relationships and, even further, their interpersonal support systems. Watt et al. found that college students with high tendencies toward boredom have lower levels of psychosocial development and poorer peer relationships than those with low boredom tendencies [26]. Therefore, it is hypothesized that social support as an extrinsic resource may mediate the relationship between boredom and meaning in life; that is, boredom affects meaning in life through social support.

To sum up, this study aims to examine the predictive effect of boredom on the two dimensions of meaning in life through a longitudinal study and to test the mediating effects of cognitive flexibility and social support between boredom and meaning in life. It expands the understanding of meaning in life and lays the theoretical foundation for an in−depth investigation of the mechanism of boredom’s influence on meaning in life.

## 2. Materials and Methods

### 2.1. Sample and Procedures 

A total of 728 university students from a Chinese university participated in this longitudinal study, in which 13 participants who did not wish to continue to participate in the survey and did not fill out all the questionnaires seriously were removed. Altogether 715 valid participants were obtained, among which 420 (58.74%) were men and 295 (41.26%) were women. The mean age was 19.48 ± 1.31 years.

Longitudinal measurements were taken at 2 time points. The first, in September, examined participants’ state of boredom; the second, in November, examined participants’ cognitive flexibility, social support, and meaning in life. A pre−trained administrator was in the classroom to organize each test, who also informed the participants that participation was completely voluntary and that they could quit at any stage of the study. Those who completed all tests were given a small gift as a token of appreciation.

### 2.2. Measures

#### 2.2.1. Multidimensional State Boredom Scale (MSBS)

The Multidimensional State Boredom Scale was developed by Fahlman et al. (2013) [27]. This study adopted its Chinese version translated and revised by Liu et al. [28], which consists of 24 items and covers 5 dimensions as follows: Disengagement (DIS), High Arousal (HA), Low Arousal (LA), Inattention (INA), as well as Time Perception (TP). The 7 point Likert scale was used from 1 (strongly disagree) to 7 (strongly agree). Higher total scores indicated a higher level of boredom. The internal consistency coefficient of the questionnaire was 0.92, and that of the sub−dimensions were 0.69, 0.81, 0.84, 0.80, and 0.83, respectively.

#### 2.2.2. Meaning in Life Questionnaire (MILQ)

The Meaning in Life Questionnaire developed by Steger et al. (2006) [10] is a brief scale that assesses two dimensions of meaning in life: the presence of and searches for meaning, which measures individuals’ tendency to pursue the value of life and the perceived purpose of and value of life, respectively. The Chinese version, translated by Liu et al. in 2010 [29], was used in this study, which contains 10 questions using the 7 point Likert scale from 1 (strongly disagree) to 7 (strongly agree), with higher scores representing higher levels of meaning in life. In this study, the internal consistency coefficient of the questionnaire was 0.80, and that of presence and search for meaning were 0.86 and 0.87, respectively.

#### 2.2.3. Cognitive Flexibility Inventory (CFI)

The Cognitive Flexibility Questionnaire was developed by Dennis et al. (2010) [18]. It contains 2 dimensions of selectivity and controllability, with 20 questions on a 5 point scale from 1 (never) to 5 (always). Higher scores indicated greater cognitive flexibility. The Chinese version of the questionnaire, translated by Wang et al., was adopted in this study [30]. The internal consistency coefficient of the as is 0.92, and that of selectivity and controllability were 0.92 and 0.85, respectively.

#### 2.2.4. Perceived Social Support Scale (PSSS) 

The Scale of Perceived Supports, developed by Blumenthal et al. (1987) [31] and revised by Jiang in 2001 [32], is a 12−item questionnaire containing three dimensions of social support with friends, family, and significant others. A 7 point Likert scale from 1 (strongly disagree) to 7 (strongly agree) is used, with higher scores indicating higher levels of perceived support. The internal consistency coefficient of this questionnaire was 0.95, and that of the three dimensions were 0.92; 0.90, and 0.89, respectively.

### 2.3. Data Analysis

This study adopted SPSS 26.0 (IBM Corp., Armonk, NY, USA) for descriptive statistics analysis and correlation analysis. Mplus 8.3 (Muthén & Muthén, Los Angeles, CA, USA) was used to construct the structural equation model.

## 3. Results

### 3.1. Descriptive Statistics and Correlation Analysis

Descriptive tests and Pearson correlation analysis were conducted concerning the five variables to obtain the means, standard deviations, and variable correlation matrices. Detailed dimensional scores are shown in Table 1 below. 

As shown in Table 1, state boredom at T1 significantly negatively correlated with the presence of meaning in life at T2 (*p* < 0.01) and did not correlate with the search for meaning in life at T2 (*p* > 0.05). The presence of meaning in life was significantly related to the search for meaning in life (*p* < 0.05). It suggests that there may be differences in the mechanisms of effects of state boredom on the two dimensions of meaning in life. State boredom at T1 significantly negatively correlated with social support (*p* < 0.01) and cognitive flexibility (*p* < 0.01) at T2.

### 3.2. Mediation Model

#### 3.2.1. Measurement Model

The measurement model included five latent variables (T1 state boredom, T2 cognitive flexibility, T2 social support, T2 the search for meaning in life, and T2 the presence of meaning in life). Results showed that the measurement fits well: CFI = 0.95, TLI = 0.94, RMSEA = 0.05, and SRMR = 0.06, indicating that a subsequent structural model analysis could be conducted.

#### 3.2.2. Structural Model

The constructed structural equation model with grade and gender being controlled as well as the presence of and search for meaning being dependent variables, is shown in Figure 1. The model fits are: CFI = 0.93, TLI = 0.92, RMSEA = 0.07, and SRMR = 0.08. Table 2 shows the standardized estimates and confidence intervals of all direct and indirect paths. The direct path between state boredom and the presence of meaning in life was not significant (β = −0.03, *p* > 0.05, 95% CI = [−0.13, 0.07]). The intrinsic resource (cognitive flexibility) could mediate between state boredom and the presence of meaning in life (β = −0.31, *p* < 0.001, 95% CI = [−0.45, −0.22]); the extrinsic resource (social support) could not mediate between state boredom and the presence of meaning in life (β = −0.03, *p* > 0.05, 95% CI = [−0.07, −0.01]). The total effect of state boredom on the presence of meaning in life is significant (−0.37, *p* < 0.001). The results indicated that state boredom played a destructive role in the presence of meaning in life.

The direct path between state boredom and the search for meaning in life was significant (β = 0.29, *p* < 0.05, 95% CI = [−0.07, 0.45]). The intrinsic resource (cognitive flexibility) could mediate between state boredom and the search for meaning in life (β = −0.23, *p* < 0.001, 95% CI = [−0.40, −0.12]); the extrinsic resource (social support) could play a mediating role between state boredom and the search for meaning in life (β = −0.13, *p* < 0.001, 95% CI = [−0.21, −0.07]). However, the total effect of state boredom on the search for meaning in life is not significant (−0.06, *p* > 0.05). The results indicated that state boredom has no significant role in the search for meaning in life.

## 4. Discussion

This study used a longitudinal survey method to explore the relationship between state boredom and meaning in life. This study examined the mediating roles of intrinsic resources (cognitive flexibility) and extrinsic resources (social support) between state boredom and the presence of meaning in life and between state boredom and the search for meaning in life, respectively. First, the findings showed that state boredom did not directly affect the presence of meaning in life. Instead, it mediated the presence of meaning in life through cognitive flexibility, i.e., the higher the level of state boredom, the lower the level of cognitive flexibility, and the lower the level of the presence of meaning in life. Although social support did not mediate between state boredom and the presence of meaning in life, the total effect of state boredom was significant, reducing an individual’s level of the presence of meaning in life in the long run. Second, state boredom had a significantly positive direct effect on the search for meaning in life, with cognitive flexibility and social support mediating between state boredom and the search for meaning in life. The higher the level of boredom, the lower the level of cognitive flexibility and social support, and the lower the search for meaning in life. However, the total effect of state boredom on the search for meaning is not significant, indicating that the overall effect of state boredom is largely insignificant.

This study added to and complemented the empirical research literature on the relationship between boredom and meaning in life, expanding the scope of related research. Unlike previous studies, which have used meaning in life as an antecedent variable, this study has used state boredom as an antecedent variable to explore the relationship between boredom and meaning in life. Most prior studies have used existentialist theories as a starting point, treating meaning in life as an antecedent variable and boredom as an outcome variable while exploring the relationship between these two variables. Although a study conducted by Fahlman et al. found a negative predictive effect of boredom on meaning in life, it did not separately examine the effect of boredom on the two dimensions of meaning in life [9]. The present study examined the role of state boredom on the presence of and the search for meaning in life as two separate entities. Indeed, when Steger et al. proposed that meaning in life encompassed both the presence of and search for meaning in life, they clarified that the two phenomena served different functions [33] and thus that the effect of boredom on each should be considered. Our study confirmed this hypothesis, finding that state boredom does, in fact, differently affect the two dimensions of meaning in life, having a negative effect on the presence of meaning in life but an insignificant overall effect on the search for meaning in life. Studies have shown that boredom triggers various negative emotional and cognitive responses, causing people to feel dissatisfied with their environments and confused about life. By reducing the sensations of a meaningful life, boredom affects people’s perception and understanding of the meaning and purpose of life [34]. This means that boredom reduces the presence of meaning in life. Although the direct effect of state boredom on the search for meaning in life is significant, the total effect is not significant, suggesting that the effect of state boredom on the search for meaning in life is complex and requires further investigation.

This study also explored the mechanisms through which state boredom acted on two dimensions of meaning in life. First, our findings confirmed that state boredom could act through intrinsic resources (cognitive flexibility) on the presence of meaning in life. In this, our findings were inconsistent with the mediating pathway of the five “A”s in the meaning maintenance model because they did not show that boredom aroused the compensatory effect of cognitive flexibility; instead, boredom actually reduced an individual’s cognitive flexibility. According to prior studies, negative emotional states are negatively associated with an individual’s cognitive flexibility [35]; this finding supported the results of the present study. Cognitive flexibility is the ability to shift between different styles of cognition and reaction in response to changes in need and context [36]. During this process, individuals must focus on the situation they are in, notice the changes that have occurred, and identify the strategies they need to adopt in response. However, individuals in a state of boredom will often show inattention, a decreased perception of time, a sense of emptiness, and negativity [37], all of which can impede cognitive flexibility. In this regard, our study did not find a mediating role of external resources (social support) between boredom and the presence of meaning in life, possibly because state boredom reduces an individual’s use of social support. Related research has shown that emotional states are linked to the expansion of interpersonal relationships, with individuals in positive emotional states typically adopting positive attitudes toward themselves and others and feeling increased tolerance and empathy for others [38]. Couples in positive emotional states were found to have more harmonious relationships and tended to use “we” more often than “I” when describing their relationships [39]. The opposite was true for individuals in negative emotional states. Prinstein et al. found that individuals with low levels of social support tended to show more negative responses, such as anxiety, and to view themselves negatively in comparison with those with higher levels of social support [40]. Social support thus serves as an important source of meaning in life. People who do not receive effective social support or feel care and respect from others may lack a sense of belonging, which in turn will affect their perception and understanding of meaning in life.

Additionally, our findings confirm that state boredom can act on the search for meaning in life through both intrinsic resources (cognitive flexibility) and extrinsic resources (social support). It is worth noting, however, that the total effect of state boredom on the search for meaning in life was not significant. Although the direct effect of boredom on the search for meaning in life was positively significant, the effects of the two indirect paths from state boredom to the search for meaning in life were negatively significant, and a suppression effect was observed. We, therefore, see the total effect of state boredom on the search for meaning in life as insignificant. This suggests that, although state boredom has the effect of motivating individuals to seek new stimuli and escape boredom [41], it can also impede the search for meaning in life by reducing cognitive flexibility and social support. Thus, overall, we do not see a significant effect of state boredom on the search for meaning in life. Further research is needed to verify this result.

In summary, the effect of state boredom on meaning in life revealed that when individuals, especially college students, fall into a state of boredom for a long period of time, boredom diminished their cognitive flexibility and social support. Individuals could not escape this dilemma by relying solely on their own ability to self−regulate. For this reason, high school and university educators should work to enhance the support provided by internal and external resources, for example, by providing college students with methods of alleviating boredom, such as mindfulness training. At the same time, educators should aim to make the learning environment more stimulating and enrich the form and content of teaching and learning, thereby increasing student engagement. Such steps could help students improve their ability to deal with boredom and thus gain more meaningful experiences.

However, it should be acknowledged there are shortcomings in this study. Firstly, in terms of the participants, although it was confirmed that boredom effected the meaning of life, the participants were confined to college students in general higher education institutions. Therefore, future studies can explore the relationship between boredom and meaning in life in other groups to advance the representativeness of the study results. Secondly, in terms of research methods, this study mainly used self−reported measures, which can be advanced in further studies by adding behavioral observations or others’ evaluations to examine individuals’ boredom and meaning in life from multiple perspectives. In the process of follow−up measurements, college students also experienced different learning stages at the beginning of the school year versus midterm with various stressful events, which might have an impact on their experience of boredom and meaning in life. Thirdly, in terms of the research contents, this study explored the mechanism of the role of boredom in influencing the meaning in life in terms of intrinsic and extrinsic resources orientation. The role of cognitive flexibility in intrinsic resources and social support in extrinsic resources are examined, while other resource factors still remain to be investigated. 

## 5. Conclusions

This study used a combination of questionnaires as well as a longitudinal design to conduct an exploratory study of the relationship between boredom and meaning in life, taking college students as research participants. This study found that boredom has different effects on the presence of meaning in life and the search of meaning in life. Cognitive flexibility played a mediating role between boredom and the presence of meaning in life, but social support did not. The total effect of boredom on the presence of meaning in life was significant. Cognitive flexibility and social support played a mediating role between boredom and the search for meaning in life, respectively, but the overall effect of boredom on the search for meaning in life was not significant. Findings revealed that individuals’ sense of meaning in life could be enhanced by reducing boredom and increasing cognitive flexibility and social support.

## Figures and Tables

**Figure 1 ijerph-19-12255-f001:**
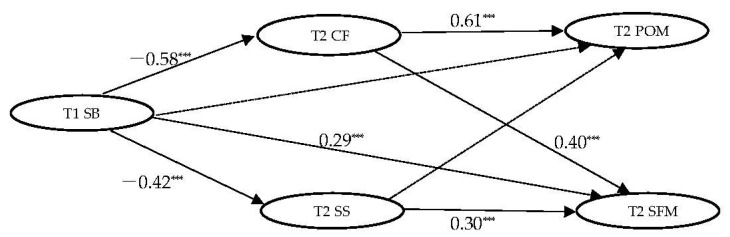
Mediating Model. *** *p* < 0.001.

**Table 1 ijerph-19-12255-t001:** Descriptive statistics and correlation analysis.

	M	SD	T1 SB	T2 SS	T2 CF	T2 SFM
T1 SB	3.06	0.90				
T2 SS	5.13	1.06	−0.34 **			
T2 CF	3.58	0.52	−0.38 ***	0.53 ***		
T2 SFM	4.90	1.04	−0.01	0.31 ***	0.33 ***	
T2 POM	4.51	1.03	−0.36 ***	0.35 **	0.53 ***	0.09 *

Note: * *p* < 0.05, ** *p* < 0.01, *** *p* < 0.001. SB = state boredom, SS = social support, CF = cognitive flexibility, SFM = the search for meaning in life, POM = the presence of meaning in life.

**Table 2 ijerph-19-12255-t002:** Mediating Effects and Confidence Intervals.

Path	β	95% CI
T1 SB→T2 SS→T2 POM	−0.03	[−0.07 0.01]
T1 SB→T2 SS→T2 SFM	−0.13 ***	[−0.21 −0.07]
T1 SB→T2 CF→T2 POM	−0.31 ***	[−0.45 −0.22]
T1 SB→T2 CF→T2 SFM	−0.23 **	[−0.40 −0.12]

Note: ** *p* < 0.01, *** *p* < 0.001. SB = state boredom, SS = social support, CF = cognitive flexibility, SFM = the search for meaning in life, POM = the presence of meaning in life.

## Data Availability

The data presented in this study are available on reasonable request from the corresponding author.

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
