# Peer review of "The Effect of Boredom on College Students’ Meaning in Life: A Longitudinal Mediation Model"

_ijerph, 2022, doi:10.3390/ijerph191912255_

Round 1
Reviewer 1 Report
The authors investigated the relationship between boredom and meaning in life and the mechanism between them through two time point measurements. The research design is reasonable and rigorous, and the research statistical method is appropriate, which can provide enlightenment for improving the meaning in life of college students. There are also some problems to be solved in the study: 1.In order to make the literature review more clear, it is suggested to add subheadings appropriately. 2. When the data less than 1,whether 0 is keep or not before the decimal point? It should be consistent. 3. In Table 1, 1 on the diagonal can be considered to be deleted. 4.The practical significance of this study should be appropriately added in the abstract, summary and discussion.
Reviewer 2 Report
The merit of this study is its longitudinal character with over 700 participants (students from a Chinese university). The content is quite important as it assesses the effect of boredom (t1) on measures such as meaning of life and search for meaning of life. Here are some suggestions for modifications (in sequence of appearance in the text):
Page 1, lines 41ff: The semantics is probably not correct. It is not that "most studies have examined the effects of meaning of life on boredom" but, probably the authors wanted to write: "In those studies that have examined the relation of meaning of life and boredom, boredom was not taken as antecedent variable."
In addition to cognitive flexibility also a related term, self-regulation capacity seems important for understanding boredom. See, for example: Witowska et al. (2020). What happens while waiting? How self-regulation affects boredom and subjective time during a real waiting situation. Acta Psychologica, 205, 103061.
In the second paragraph of 2.1. Sample and Procedures, change the present tense to past tense.
Give information how questionnaires were presented. In presence or in an online study?
The Multidimensional State Boredom Questionnaire (MSBS): Is it not that the MSBD is used to capture boredom after an event? Or how are the questions asked. The participants were either in a class room (presence) or at a computer (online study) when they filled out the MSBS. To what did the MSBS questions refer. "State" means feeling now.
Table 1 is at the wrong position. It is part of results but appears in the Methods section. Also, in the Table caption write whar T1,T2,SS,SB, etc. mean.
Delete the first three lines of the results section ...
Even though the authors are later conducting a mediation analysis, the correlations (and Table 1) are confusing because one would anticipate a regression analysis for t1 => t2 and correlations between variables at T2.
There is the indication of Table 3.1 and Figure 3.2 where they should be named Table 1 (line 183) and Figure 2 (line 200).
Page 4, line 185: Replace: "correlates insignificantly" with "does not correlate".
A wrong "comma" in the brackets in line 186.
The Discussion needs some more thorough editing. First of all, the English has lower quality in the Discussion than elsewhere. E.g. "effect on presence of life" (add: meaning); "living in the state boredom", ...
Then the Discussion does not seperate clearly between what is found in the present study and what is knowledge from past studies. Make clearer. many parts of the Discussion read like the repitition of the Introduction. But now we know more. For example, the Discussion should start with a paragraph on "we found 1., 2., 3., and 1. is new, 2. contradicts former studies, 3. complements former studies. What do we learn from the new study results?
Author Response
We would like to express our gratefulness to the valuable and constructive comments and suggestions to help improve the quality of our research work. The responses to the reviewers’ comments point by point are provided as follows:
Point 1: Page 1, lines 41ff: The semantics is probably not correct. It is not that "most studies have examined the effects of meaning of life on boredom" but, probably the authors wanted to write: "In those studies that have examined the relation of meaning of life and boredom, boredom was not taken as antecedent variable."
Response 1: Thank you for your suggestion. We did not give the correct expression. What you propose to change is exactly what we want to write. So, we have changed the sentences where lines 45-47 to " In most studies that have examined the relation of meaning of life and boredom, boredom was not taken as antecedent variable.”
Point 2: In addition to cognitive flexibility also a related term, self-regulation capacity seems important for understanding boredom. See, for example: Witowska et al. (2020). What happens while waiting? How self-regulation affects boredom and subjective time during a real waiting situation. Acta Psychologica, 205, 103061.
Response 2: Thank you for your advice. We have read the paper you recommended, which discussed the process of self-regulation and showed that self-regulation may be an important factor in boredom. Cognitive flexibility is the human ability to adapt one’s cognitive processing strategies to face new and unexpected conditions. From this perspective, cognitive flexibility may be also a reflection of individual self-regulation ability. Of course, self-regulation also involves the regulation of emotions. However, in this manuscript, the role of emotion regulation between boredom and sense of meaning in life is not studied, and future research can further explore it.
Point 3:In the second paragraph of 2.1. Sample and Procedures, change the present tense to past tense.
Response 3: Thank you for your suggestion. We have changed the present tense to past tense in Sample and Procedures. It can be found in line134-141.
Point 4: Give information how questionnaires were presented. In presence or in an online study?
Response 4 : Participants were in the classroom, and pre-trained experimenters handed out questionnaires, which were then filled out by participants.
Point 5:The Multidimensional State Boredom Questionnaire (MSBS): Is it not that the MSBS is used to capture boredom after an event? Or how are the questions asked. The participants were either in a class room (presence) or at a computer (online study) when they filled out the MSBS. To what did the MSBS questions refer. "State" means feeling now.
Response 5: The Multidimensional State Boredom Scale (MSBS) is used to assess the level of boredom of individuals in general situations. This scale is not necessarily meant to measure boredom after an event.
The MSBS contains five factors: 1. Disengagement(“I seem to be forced to do things that have no value to me”);2. High Arousal (“I feel agitate”“I am annoyed with the people around me”); 3. Low Arousal (“I feel down””I feel empty”); 4. Inattention(“I am easily distracted”); 5. Time Perception(“Time is passing by slower than usual””Time is dragging on”).
The participants were asked to filled out the MSBS by how they feel right now in a class room.
Point 6:Table 1 is at the wrong position. It is part of results but appears in the Methods section. Also, in the Table caption write what T1,T2,SS,SB, etc. mean.
Response 6: Thank you for your suggestion. The table 1 has been put in the result section. Please refer to lines 183 to 187 for details of modification
Point 7:Delete the first three lines of the results section ...
Response 7: Thank you for your suggestion. But we didn’t find the first three lines of the results section. It is not quite sure if this issue is caused by the computer's display version.
Point 8: Even though the authors are later conducting a mediation analysis, the correlations (and Table 1) are confusing because one would anticipate a regression analysis for t1 => t2 and correlations between variables at T2.
Response 8: This confusing situation occurs, it may be that the content of the previous table 1 is not standardized, and now it has been adjusted. Please refer to lines 183 to 184 for details of modification
|
|
M. |
SD. |
T1 SB |
T2 SS |
T2 CF |
T2SFM |
|
T1 SB |
3.06 |
0.90 |
|
|
|
|
|
T2 SS |
5.13 |
1.06 |
-0.34** |
|
|
|
|
T2 CF |
3.58 |
0.52 |
-0.38*** |
0.53*** |
|
|
|
T2 SFM |
4.90 |
1.04 |
-0.01 |
0.31*** |
0.33*** |
|
|
T2POM |
4.51 |
1.03 |
-0.36*** |
0.35** |
0.53*** |
0.09* |
Point 9:There is the indication of Table 3.1 and Figure 3.2 where they should be named Table 1 (line 183) and Figure 2 (line 200).
Response 9: Thank you for your suggestion. Table 3.1 and Figure 3.2 have been revised to Table 1 and Figure 1 as you suggested. Please refer to line 183 and line 224 for details of modification
Point 10 : Page 4, line 185: Replace: "correlates insignificantly" with "does not correlate".
Response 10: Thank you for your advice. We have replaced "correlates insignificantly" with "does not correlate" in line 189.
Point 11: A wrong "comma" in the brackets in line 186.
Response 11: A wrong "comma" in the brackets in line 186 has been deleted.
Point 12:The Discussion needs some more thorough editing. First of all, the English has lower quality in the Discussion than elsewhere. E.g. "effect on presence of life" (add: meaning); "living in the state boredom", ...
Response 12: Thank you for your suggestion. We asked professional translators to polish the discussion to make the English writing more accurate.
Point 13:Then the Discussion does not seperate clearly between what is found in the present study and what is knowledge from past studies. Make clearer. many parts of the Discussion read like the repitition of the Introduction. But now we know more. For example, the Discussion should start with a paragraph on "we found 1., 2., 3., and 1. is new, 2. contradicts former studies, 3. complements former studies. What do we learn from the new study results?
Response 13: Thank you for your suggestion. The discussion section has all been adjusted and revised according to your comment. First , we reported the study result. Second, we reported the new findings: 1. state boredom as an antecedent variable has different effects on two dimensions of meaning in life separately. 2. cognitive flexibility played a mediating role between boredom and the presence of meaning in life, but social support did not.The total effect of boredom on the presence of meaning in life was significant. 3. cognitive flexibility and social support played a mediating role between boredom and the search for meaning in life, respectively, but the overall effect of boredom on the search for meaning in life was not significant. Third, according to the results of the study, we discussed that that how to help colleges students to reduce boredom and improve the sense of meaning life by enhance the support provided by internal and external resources.
In addition, we found that the content of the analysis of the results was not detailed enough in the process of revising the manuscript. We supplemented path coefficient between the variables in the structural equation. Please refer to line 205 to 222 . Then, we revised the content of the abstract, discussion and conclusion accordingly.
P.S. Please see attachment for the revised manuscript
